# Biobanks—A Platform for Scientific and Biomedical Research

**DOI:** 10.3390/diagnostics10070485

**Published:** 2020-07-16

**Authors:** Kristina Malsagova, Artur Kopylov, Alexander Stepanov, Tatyana Butkova, Alexandra Sinitsyna, Alexander Izotov, Anna Kaysheva

**Affiliations:** Biobanking Group, Branch of IBMC “Scientific and Education Center” Bolshoy Nikolovorobinsky lane, 7, 109028 Moscow, Russia; a.t.kopylov@gmail.com (A.K.); aleks.a.stepanov@gmail.com (A.S.); t.butkova@gmail.com (T.B.); anvilya@gmail.com (A.S.); farmsale@yandex.ru (A.I.); kaysheva1@gmail.com (A.K.)

**Keywords:** biobank, bioethics, human samples, personalized medicine, digitization

## Abstract

The development of biomedical science requires the creation of biological material collections that allow for the search and discovery of biomarkers for pathological conditions, the identification of new therapeutic targets, and the validation of these findings in samples from patients and healthy people. Over the past decades, the importance and need for biobanks have increased considerably. Large national and international biorepositories have replaced small collections of biological samples. The aim of this work is to provide a basic understanding of biobanks and an overview of how biobanks have become essential structures in modern biomedical research.

## 1. Introduction

Collections of biological material have been directly related to the development of biology as a science. Biomaterial collections date back to the 17th century. One of the most famous examples of biological collections is the Carl Linnaeus Botanical Collection, which was created not only to solve the scientific problems of plant classification but also for the practical tasks performed in botanical gardens [1]. Today, various synonyms are used interchangeably—biological storage, biological repository, biological collection. The term “biobank” can be referred to an individual collection, which is stored in a private refrigerator of a particular research group within a specific project. Alternatively, the UK national biobank (https://www.ukbiobank.ac.uk/), which contains several million samples and has a highly centralized approach to collecting, processing, and storing samples, is also categorized as such. The term “biobank” usually refers to a large, organized collection of well-characterized tissue samples, such as surgical biopsy specimens—freshly frozen or in paraffin sections, blood and serum samples, various types of cells and DNA—all carefully collected for research purposes and annotated [2]. The Public Population Project in Genomics (P3G, http://www.p3g.org/) defines biobanks as an infrastructure for collecting human biological samples and related data, such as medical, family, social, and genetic [3].

Large international communities, such as the Organization for Economic Co-operation and Development (OECD), the International Community for Biological and Environmental Repositories (ISBER), the European Commission (EC) have their own definitions for the term “biobank”. Therefore, for example, the OECD defines a biobank as a collection of biological material, related data and related information stored in an organized system for the needs of an entire population or part of it [4].

ISBER views the biobank as an organization that receives, stores, processes and/or distributes biological samples in an appropriate manner. That is, it is engaged in the physical placement of samples and the full amount of work associated with them [5].

The European Commission also gives its own definition for biobank as an organized collection consisting of biological samples and related data that are of particular importance for fundamental science and the needs of personalized medicine [6].

Several general characteristics hold true for any biobank. First, there is the mandatory storage of biomaterial, including, but not limited to, cells, DNA structures, plasma and/or serum, tissue, etc. The origin of biological material may vary from humans, plants, and animals to microorganisms. Some biobanks contain collections from different biological species. However, collections of human biomaterials are of particular interest.

The biobank is a new institution created at the intersection of science, technology, and business. The scientific component of biobanks is associated not only with the storage of biological samples and the extraction of information from them, but also with the formulation and solution of a wide range of problems related to the description of biodiversity, the identification of evolutionary patterns in living organisms on Earth, and the identification of prospects within evolutionary processes. The technological component of biobanks is related to the fact that any biobank has a certain infrastructure. For example, cryopreservation of biological samples requires specialized storage methods utilized for long-term storage [7].

The information that is extracted from biological material also requires informational resources to support the infrastructure [8]. These are large databases that not only necessitate a constant update but are also used in comparative studies within various fields. In addition, biobanks deal with the use of results of scientific and technological activities in commercial fields. For example, pharmaceutical biobanks are developing at the fastest pace, as pharmaceutical companies are interested in accelerating and reducing the cost of research at the preclinical stage of drug testing by utilizing genetic material [8]. Biobank resources can reveal common pathologies with varying frequencies in populations, allowing for pharmaceutical companies to allocate capital for drug development in health sectors and research areas where demand is predicted. Therefore, the biobank is an institution of a new type, combining the interests of various stakeholders which often do not coincide.

Biobanks not only accelerate the process of transferring scientific knowledge into technological applications, but also give rise to situations where the process of obtaining scientific knowledge is largely dependent on technology. That is, the interaction of scientific and technological processes is changing as dependences can now be observed not only from science to technology but also from technology to science.

In the field of psychogenetics, the use of the genealogical method and population-based studies in general allow us to evaluate the contribution of genes and environment to the formation of personality, and biobank collections bring forth a new dimension to the field [9]. Biobanks can develop only if there is substantial public support. Accordingly, the willingness and awareness of citizens, who are ready to provide their biomaterials for scientific and applied research to biobanks, depends to a large extent on the psychological readiness to alienate one’s own biomaterial for an altruistic contribution to the development of science. Further, it is of relevance to mention the question of possible compensation for the use of one’s genetic material in the event of a particular private corporation recognizing an opportunity to profit from the resulting knowledge and its processing.

An analysis of the literature on research related to the organization or development of biobanks was carried out in March 2020 using the PubMed database. Our search criteria used the words “biobank” or “biobanking”. Over the past five years, the number of published scientific articles has amounted to more than 6000. There has been a 4-fold increase in the number of articles since 2015.

The development of omics (genomics, transcriptomics, proteomics, metabolomics) has contributed to progress in the field of biobanking and improved the conduct of biomedical research. As a result, large electronic databases have appeared that store enormous amounts of information (big data) associated with clinics [10]. Thus, biobanks play a key role in the era of precision medicine, which is based on the analysis of fully annotated samples. Having a diverse collection of patient samples (with well-annotated clinical and pathological patient data) is a critical requirement for personalized medicine.

Scientific programs, designed for the longitudinal collection of accessible biological fluids from the person over their lifetime, are of particular interest [11]. In this case, there is a unique opportunity for usage of your own biological substance as a control basement for the diagnostics. Thus, the issue of inter-individual variations in plasma and urine biomarker levels can be eliminated [12]. Currently, the foundation of biobanks in combination with patients register is considered as the main source for conducting translational research on the implementation of fundamental knowledge in practical medicine [13]. Simpler blood plasma banks, DNA preparations [14], as well as tissue banks [15] that require more sophisticated equipment and standard operating procedures, allow the scientific community to investigate and discover for new diagnostic and prognostic markers of diseases, to segregate nosological types into molecular subtypes, to provide molecular epidemiological studies, including retrospective manner, taking into account the disease outcome [16].

If more high-quality samples are made available through biobanks, researchers can use these resources to improve patient care [17].

In this review, biobanking organization is considered as a promising area in the field of biomedical research, as a key resource for improving the quality of scientific research and participation in global research. This review purposed to cover the concept of modern biobank including aspects of ethical–legal issues, data safety and appropriate operating procedures specified for personalized medicine and bespoke treatment.

## 2. Organization of a Biobank

Biobanks are heterogeneous in their design and use, as they may contain data and samples obtained as a result of family studies or from patients with a certain disease, as well as control groups. Biobanks may also be generated from large-scale epidemiological research or collections from clinical trials of new medical interventions.

Human biological samples include a wide range of tissues, organs, body parts, isolated DNA or RNA, blood, plasma, other biological fluids, cell lines, cell suspensions, etc. Currently, a large number of research projects are focused on the areas of genomics, transcriptomics, proteomics, and metabolomics, and study tissue samples from patients with established clinical and pathological diagnoses or use cell lines/suspensions obtained from the blood of patients.

The standardization and centralization of the processing and sampling of biosamples provides a cost reduction as well as increased throughput and accuracy. For example, this can be achieved using automation. The influence of analytical variability will also be limited, thereby improving the prospects of a subsequent study, which would use data from the previously analyzed biosamples. An undetectable systematic error introduced by variable (usually manual) processing of several objects should be avoided. Given that biobank resources are designed for the development of diagnostics, prognostics, prevention, and studying of the etiology of complex diseases in which exposure to specific risk factors is typically low (odds ratio is usually 1.5 or lower), this type of error can cause inaccurate results or mask the presence of a real causal factor. Such an effect may be aggravated in future cohorts where case-control studies are included in the sample, especially if the biomaterial was obtained according to different protocols performed at different clinical bases. If the production of biomaterial and its processing takes place within different premises, then considerable efforts should be directed at training personnel in agreed and approved operational procedures. Further, the monitoring of effectiveness should be carried out in order to ensure compliance with quality standards. Only the responsible and high-quality application of standard operating procedures within the framework of individual studies can allow for biobanks to be used for effective cooperation and the creation of a very large “virtual” sample size at the international level [18].

The controlled, highly qualified organization and operation of storage facilities and/or infrastructures for biomedical research is the ultimate goal of biobanking. For this reason, the new ISO/Technical Committee (TC) 2761/Working Group (WG) 2 Biobanks and Bioresources was organized in 2014 to incorporate the new ISO/DIS 20387 standards in biobanking area that guides handling and operating procedures with animal, plant, microorganism and human sources [19,20,21]. Compliance with these requirements will enable biobanks to work consistently, professionally, and provide biological resources of adequate quality.

ISBER is a leading global platform that focuses on the development, management and use of storage facilities. One of the key goals of ISBER is the exchange of successful strategies, rules and procedures for the provision of quality samples for research. Document [22] provides good practices for managing collections and sample repositories.

In addition, an agreement is necessary to regulate the transfer of biomaterial between biobanks and research groups, to maintain certain quality standards, and to register the routing of biological samples [23].

One of the models of the working process for the collection, storage, and distribution of biological samples in biobanking is presented in Figure 1.

After the collection of biological material and compliance with the common ethical standards, biological samples are marked and recorded in accordance with the clinical information in the electronic data capture system. Before cryopreservation, the biomaterial is aliquoted, preferably using robotic systems. The biobank managing software must have the following characteristics: (1) ability for specimens’ management including, but not limited to, the administration of consents, history of biological sources, non-conformity management, storage and delivery, etc.; (2) tracking for the complete chain of custody that covers samples annotation and request history, clinical and genetic data; and, eventually, (3) compatibility of interface that permits monitoring the system and easy joining with collaborating laboratories and hospitals to safely extract the necessary clinical data.

## 3. Pre-Analytical Sample Handling

In biomedical research, it is becoming increasingly important to expand the analysis from individual molecules to the signatures of nucleic acids, proteins and metabolites in various biological samples of humans. However, the profiles of these molecules can change during collection, storage, and transportation. These factors lead to unreliable results, since the result does not reflect the condition in vitro, but rather an artificial profile shaped during the preliminary preparation of the sample. Therefore, the standardization of the pre-analytical stage in the structure of biobanks is mandatory, since variable factors at the stage can produce a significant impact on the reliability of analytical results [24].

The pre-analytic stage includes the following three phases:The phase preceding the biological material collection. Here, factors related to the subject (genotype, lifestyle, nutrition, medication, concomitant diseases, surgical interventions, etc.), as well as conditions of samples obtaining (for example, on an empty stomach or postprandial), are of great importance. Most of these variables cannot be standardized because they relate to individual subject and specific treatment methods. However, all of these parameters may affect subsequent analytical results of research and diagnosis.The collection phase of biological material begins when the biobank’s staff receive samples. At the moment, the state of the samples is influenced by conditions, transportation (compliance with the cold chain regime, duration) and chain of custody in the laboratory.The phase after collecting includes registering and proper annotation of samples in the biobank database. Different storage conditions and time, control of the archived samples, procedures for isolating of analytes for subsequent research can influence the total outcome of studies.

For phases 2 and 3, standardization is more accessible since evidence-based best practice protocols can be applied and implemented.

There are many common elements to sample collection and retrieval practices, while sample processing methods differ depending on the particular study or clinical activity associated with the repository. Prior to sampling, the accessibility of samples and potential analytical purposes for their use should be assessed, and methods should be used to verify that all collected samples are suitable for use. Using the developed ISBER best practices will minimize the errors of the pre-analytic phase [22].

One of the important activities in this area is Standardization and improvement of generic pre-analytical tools and procedures for in vitro diagnostics (SPIDIA) [25]. This is a large-scale European project in collaboration with representatives of the industrial sector and scientific communities, designed to study the influence of critical pre-analytical variables of biological samples in molecular analysis.

The requirements for quality control, standardization of the pre-analytical stage and subsequent standard operating procedures in biobanks depend on the type of sample and the analytical platform to be used. For example, the metabolic profile can be considered as the final response of the human being to environmental changes, particularly in the field of nutrition research, since metabolites are the end products of cellular regulatory processes. Their importance in biomedical research, including the detection of biomarkers and future clinical applications, depends on how much the recovered metabolic state actually reflects an in vitro metabolic state. The metabolome can be perceived to have a number of variables in the pre-analytical stage, which affects the enzymatic activity, rate and outcome of chemical reactions or apoptotic processes before and during the collection, transportation of the sample and storage. Therefore, the effect of pre-analytical stage should be evaluated, monitored and, if possible, diminished. Recently it has been shown that the metabolic profile of human liver tissue samples during ischemic disease and after surgery was significantly dissimilar [26]. Such studies are of great importance for biobanks as a part of biomarker studies, so researchers should be aware of the effects of pre-analytics and analytical platforms they routinely working with. The European Committee for Standardization regularly conducts numerous activities based on the results of different initiatives (including SPIDIA) to develop new technical specifications for International Organization for Standardization (ISO), scopes for the standardization of pre-analytical workflow, best transfer procedures and quality assurance in molecular analysis of different types of samples.

The ISO published a series of technical specifications for molecular in vitro diagnostic studies and preliminary preparation processes, which are relevant for both diagnostic laboratories and biobanks (Table 1).

In addition, ISO developed standards for biobanks and bio-resources. The ISO 276 technical committee (standardization in the field of biotechnology) includes:Terms and definitions in the field of biobank;Biobanks and biological resources;Analytical methods;Bioprocessing;Data processing, including annotations, analysis, validation, comparability and data integration [35].

Therefore, biobanks should implement recommendations that define quality assurance (QA) and quality control (QC) for sampling, processing, storage and disposal, as well as in relation to equipment maintenance and repair, staff training, data management and record keeping, and adherence to good laboratory practice. All biobank handlings should be subject to regular audits. The timing, scope and results of these reviews should be documented.

Thus, new developments in analytical techniques and a better understanding of the effects of pre-analysis, standardization and optimization of the workflow open up new opportunities for research, but also require the introduction of new standard operating procedures for biobanking in the framework of sample collection, annotation, pre-processing, storage conditions and quality control.

## 4. Collection and Storage of Biomaterial

Blood, urine, and tumor tissues are typical biological samples in biobanks that participants provide in accordance with a certain form of informed consent and are suitable for biomedical research. However, in addition, a wider range of types of biological samples, including hair, nails, saliva, feces, microbiome and breast milk, can also be collected and processed in accordance with the guidelines [36,37,38,39,40].

Biobanks are diverse in terms of the purpose for which they collect biological material. Genomic biobanks of entire populations, or population biobanks, are large-scale repositories for biomaterial and related clinical data. A population biobank is designed to link biomarkers with a medical history and lifestyle information. Diseases resulting from mutations of a single gene are rare [41]; hence, this link provides a powerful tool that contribute to our knowledge about both the genetic and environmental determinants that entail to Alzheimers disease, asthma, diabetes mellitus, schizophrenia, cancer, and adverse outcomes such as preterm birth and congenital birth defects.

The European Commission has identified population biobanks as “a collection of biological materials that has the following characteristics: the collection has a population basis; the collection was created or was transformed to provide biological materials or data obtained from them for several future research projects; the collection contains biological materials and related personal data that may include or be associated with genealogical, medical and lifestyle data and which may be regularly updated; biobank receives and delivers materials in an organized manner” [42].

These large-scale repositories were created to collect, analyze, and store phenotypic and genetic information from representative samples of their original populations. Virtual biobanks are organized to help researchers locate biological samples for testing and collecting data from several biobanks in disparate locations. These virtual biobanks are accessed using specialized software or web portals designed to connect biobanks and researchers around the world [43].

Designed sampling protocols should be available and stored in a biobank database. The protocol features are determined by the type of biological material and the anticoagulants used.

Since blood sampling can occur in places remote from the cryogenic storage, special precautions should be taken during temporary storage and transportation. Otherwise, this can have dire consequences for the prognostic value of the identified biomarker.

### 4.1. Tissue Biobanks 

Tissue samples are usually obtained during surgery or autopsy and undergo histological examination immediately thereafter. At this stage, the most appropriate clinical practice for preventing tissue degradation and reducing undesired enzymatic activity is tissue fixation, which is usually done using neutral buffered formalin. However, nucleic acids and proteins retain their integrity and do not lose antigenicity in fresh or frozen tissues and are, therefore, the most suitable for mass spectrometric analysis, ELISA (proteins) or amplification and sequencing (nucleic acids) [44,45,46].

Compliance with temperature conditions during the collection and maintenance of the tissue biobank are key points, the violation of which can affect the heterogeneity of the data. The standard temperature for storing tissues and cells is from −80 to −150 °C. Ultra-low temperatures maintain the integrity of proteins, DNA, RNA, and other cellular components. A temperature of −80 °C is the current standard for the preservation of human tissues/cells, although some authors recommend liquid nitrogen due to the risk of contamination with floating tissue fragments [47].

Freezing can damage living cells and tissues. Thus, cryoprotectants, such as dimethyl sulfoxide, are usually used to prevent cryoprosthesis. It is important to note that the concentration of the cryopreservation agent must be optimized depending on the type of cell or tissue in order to obtain biomaterial after defrosting while maintaining its original properties [48]. This is of particular importance when freezing bulk tissues, where the uniform distribution of the cryoprotective agent is not guaranteed due to various effects of heat and mass transfer during cryopreservation [48,49]. Conventional cryopreservation media contain fetal bovine serum, consisting of a mixture of growth factors, cytokines, and other substances, which renders its use unacceptable when establishing a standardized protocol for cryopreservation [50].

### 4.2. Blood Biobanks

Blood is one of the most common biological materials used in research. It is collected in test tubes containing preservatives and additives, depending on the specific intended use and the required blood fraction (serum, plasma, white blood cells, red blood cells). While serum samples are usually obtained by using tubes containing a coagulation activator, such as thrombin or silicon dioxide, plasma samples are obtained using tubes containing various anticoagulant additives. Most biochemical analyses are performed using serum, while plasma is used for the analysis of DNA and RNA. For example, citrate-stabilized blood contains higher-quality DNA and RNA and a higher number of lymphocytes compared to blood treated with other anticoagulants [51], while collection tubes coated with ethylenediaminetetraacetic acid (EDTA) are preferred for protein analysis and most DNA testing assays [52]. Further, heparin is suitable for metabolic studies but is not indicated for the study of lymphocytes, since it affects the proliferation of T cells [51].

The timing of the pre-analytical phase is also of considerable importance. For example, the integrity of protein molecules is maintained if blood plasma is prepared immediately [53,54], and the optimal quality of DNA extracted from blood samples will be ensured if sample preparation is carried out within 24 h at 4 °C [55]. The optimal temperature for storing blood components varies depending on the particular analyte, marker, or other molecule of interest.

As a rule, low (−20 °C) and ultra-low temperatures (−80 °C) for short- and long-term storage, respectively, are optimal for maintaining the integrity and stability of blood components [53,56].

Molecular analysis strictly depends on the methods of collection, isolation, and storage of DNA and RNA molecules. RNA is considered the most labile molecule, so the pre-analytical stage is critical. Depending on the type of biomaterial, the quantity and quality of RNA will be different. Freshly frozen tissue is an ideal sample for RNA extraction. When using formaldehyde-fixed paraffin-embedded samples, the genetic material in tissue is reduced due to formalin-induced cross-linking of nucleic acids as well as due to the time interval between tissue resection and fixation [57]. To maintain good RNA quality, samples should be stored at −80 °C without repeated freeze–thaw cycles.

DNA is more stable than RNA and can be stored at 4 °C for several weeks. However, if DNA extraction cannot be performed immediately, blood samples should be stored at −80 °C [58].

### 4.3. Cell Biobanks

The development of biological and biomedical research, which requires a large amount of diverse cellular material, as well as that of technologies and their automation in the field of cell line cultivation, have contributed to the creation of cell biobanks. It is reported that about 15% of all cell lines used in the global scientific community are incorrectly identified [59]. As a result, researchers report and read incorrect (irreproducible) data in scientific papers. This leads to a misallocation of resources for scientific research. Therefore, deposit standards and characteristics of cell lines should be universally established to provide consumers with reliable information about the origin and quality of each cell culture. These can include approved forms of cell line passports, the development of regulations for cell line collections, standard operating procedures, guidelines for expanding, maintaining the collection, working with collection materials, issuing samples, creation of a modern deposit infrastructure, and developing the material and technical base.

Currently, major cell line repositories include American Type Culture Collection, Leibniz Institute DSMZ-German Collection of Microorganisms and Cell Cultures GmbH, the European Collection of Authenticated Cell Cultures, the Japanese Cancer Research Resources Bank, RIKEN BioResource Center (Japan), and the Korean Cell Line Bank.

In Russia, a collection of cell cultures is stored at the Koltzov Institute of Developmental Biology of Russian Academy of Sciences. Many cell lines are unique (for example, induced pluripotent stem cells obtained from patients with Down syndrome), while the collection also contains a number of well-known constant lines (HaCaT, HeLa), which are in demand for various activities in the field of cell biology, regenerative medicine, and molecular biology. Currently, joint work is being carried out between the Institute of Cytology of the Russian Academy of Sciences and the Institute of Cytology and Genetics of the SB RAS to create and enrich a single All-Russian electronic catalog of cell lines.

These repositories guarantee authentic and strictly controlled in vitro model systems for medical research with appropriate certification, including disease-related data, karyotyping, immunoprofiles, unique molecular or genetic changes, growing conditions, and mycoplasma testing [60].

### 4.4. Organoid Biobanks

Recent research in stem cells and genomics has allowed the cultivation of mini organs (organoids). Organoids are self-organizing, three-dimensional structures that are similar in morphology and function to real organs, and therefore can be widely used in biomedical research and in the future in the clinic [61].

Organoids can be grown from several types of stem cells, including induced pluripotent stem cells, human embryonic stem cells and adult stem cells for a wide range of organs (intestines, kidneys, pancreas, liver, brain, etc.). Organoids can be stored in biobanks and used for basic research, accurate and regenerative medicine [62,63] including research in the field of organ development, their transplantation of drug testing [63,64].

Van de Wetering M. in his work described the creation of a living organoid biobank of colorectal cancer patients. Surgically resected tissue was obtained from patients with previously untreated colorectal cancer. Tissue from patients with colorectal cancer was excluded, as they often undergo radiation therapy before surgery. A total of 22 tumor organoid cultures and 19 normal adjacent organoid cultures were obtained. Tumor organoids are susceptible to high-throughput drug screening to detect gene associations with drugs. Therefore, organoid technology can complement studies of drugs based on the cell line and xenograft, as well as provide the possibility of an individual approach when prescribing therapy [65].

Sachs N. has developed a protocol for the long-term cultivation of human mammary epithelial organoids. Using this protocol, more than 100 primary and metastatic lines of organelles were obtained that reflect the diversity of the disease. Breast cancer organoids allowed for in vitro screening, consistent with in vivo xenograft and patient response. This study describes a representative collection of well-characterized breast cancer organoids available for research on this pathology and drug development, as well as a personalized strategy for assessing the in vitro response of a drug [66].

### 4.5. Imaging Biobanks

The development of high-performance computing systems has led to the growth of image biobanks and allows one to extract numerous quantitative characteristics from images obtained using modern scanning technologies such as computed tomography, magnetic resonance imaging, and positron emission tomography [67,68]. This research area is called “radiomics” and focuses on the validation of novel imaging biomarkers (a new class of non-invasive biomarkers) for assessing physiological or pathological processes as well as therapeutic treatment [69,70].

For example, when monitoring the condition of a patient with an oncological pathology, imaging functions are used for the measurement of tumor volume, perfusion degree, texture analysis, etc. Imaging biomarkers are non-invasive and can, in the future, replace invasive procedures such as biopsies. Imaging biomarkers are usually defined as the expression of biosignals extracted from an electromagnetic, photon, or acoustic signal obtained by monitoring a patient’s body. Imaging biobanks should include data, metadata, biomarkers obtained by image analysis [71], as well as other data related to the pathological condition (patient prognosis, pathological data, genomic profiling, etc.).

The US-based Cancer Imaging Archive (TCIA) is a service that stores medical images of cancer patients in a large archive. Digital Imaging and Communication in Medicine is the main file format used by TCIA for storing images. If necessary, additional clinical, therapeutic, and laboratory data can be provided [72].

Imaging biobanks are infrastructures with huge storage and computing capabilities. High-performance computing resources are needed to facilitate the comparison, standardization, and validation of processed images. The integration of resources and services through a platform that controls the flow of information and image processing is a necessary step for the development of image biobanks [73].

### 4.6. Digital Biobanks

Currently, digitalization covers almost all areas of modern society, including biomedical research. Digitalization facilitates the integration of data obtained from biosamples (e.g., “omics” data) with a wide range of phenotypic data obtained from other specialized research institutions or provided by patients themselves [73]. Moreover, it has become apparent that the data related to a sample must be primarily in a digital format in order to be stored in databases.

A Digital Biobank is being compiled by the Institute of Biomedical Chemistry’s Center for Collective Use “Human Proteome”. To date, the collection contains about 2000 biological samples, including serum, blood plasma, tissue biopsies, and urine samples obtained from patients with various pathologies, including oncological diseases, mental disorders, as well as samples from healthy volunteers. In addition to the standard annotation, for most samples, the metabolic-proteomic composition was studied, including quantitative measurements of the molecular composition.

The goal of the Digital Biobank is to ensure the availability of qualitatively annotated biological samples for planning research programs, an innovative and personalized approach to disease treatment and diagnosis.

In parallel to the formation of the Digital Biobank, standard operating procedures are developed and tested to ensure controlled and high-quality protocols for the collection, storage, and processing of samples. All biosamples are stored at extremely low temperatures (−80 °C) to ensure long-term stability and integrity. Samples are pre-packaged in several aliquots and then placed in ultra-low temperature conditions to prevent thawing.

The Digital Biobank system allows for the audit and inventorization of biological samples. Blood sampling is carried out using vacutainers. Tubes with sodium citrate are used for obtaining plasma, and tubes with a clot activator (SST) are used for obtaining serum. After blood sampling, the vacutainers are inverted 4–6 times to mix the anticoagulant/preservative with whole blood.

For urine collection, 15-mL plastic laboratory tubes are used, which are subsequently aliquoted (0.5–1.5 mL volume), marked with a unique identification number, and placed in temporary storage at −80 °C. The tissue samples are usually obtained during operations. Thereafter, biopsy specimens are immediately transferred to ultra-low temperature conditions. If there is no possibility for immediate freezing, neutral buffered formalin is used to prevent tissue degradation. Tissue biopsies are collected in Eppendorf tubes, assigned a unique identification number, and temporarily stored at −80 °C.

Blood in a plasma separation tube is immediately centrifuged at 2500× *g* for 10 min at 25 °C. Blood in the serum separation tube is allowed to thicken for 25–30 min at room temperature before centrifugation at 2000× *g* for 10 min at 25 °C. The prepared plasma/blood serum is then aliquoted (volume 0.5–1.5 mL), a unique identification number is marked, and the sample is placed in temporary storage at −80 °C. Thereafter, the biosamples from the temporary storage are transported to the biobank in temperature-controlled containers.

Digital biobank provides a new opportunity to annotate samples. The digital bank provides the researcher with information about the clinical parameters of the biosample (anthropometric, histochemical, biochemical), as well as molecular data—identified genetic polymorphisms, especially transcriptome, proteomic and metabolic profiles.

It is expected that the digital biobank resources will be used to assess the relevance of case-control retrospective studies. In addition, the wide availability of digital information will facilitate the generation and use of metadata in biomedical research, including, for example, the types of analysis or equipment used for a particular analysis or measurement. In addition, metadata repositories can also be used to develop common data ontologies or to harmonize laboratory measurement conditions. In the biobanking business, taking into account the high cost of creating and maintaining biomaterial collections and the rather narrow specialization of individual collections, international cooperation plays a big role. Therefore, the concept of digital network biobanks using standard data exchange formats is very promising, which will improve cooperation between different research institutes [73].

## 5. The Heterogeneity of Biobanks

The high level of heterogeneity among biobanks is primarily due to various national directives (data protection rules) [74], as well as individual approaches to data collection, storage, and annotation [69]. Data and samples are inevitably collected under various conditions for different purposes using different standards.

The stages of organizing a biobank, such as collection, identification, long-term storage, quality control, transportation, and disposal of biomaterial, are the main sources of heterogeneity. For example, tissue freezing can affect the heterogeneity and decentralization of biobanking. Surgical resection tissue usually remains at room temperature for some time until stabilized. This time window can be critical and may subsequently contribute to the degradation of the sample. Some biobanks send an employee to the operating room with a container of liquid nitrogen to reduce the time between surgery and sample stabilization, but this practice is not widely used. The storage of tissue at a low temperature immediately after surgery is an effective way to obtain high-quality biomaterial and, therefore, an additional step for standardization. The temperature conditions during collection and maintenance are also pre-analytical aspects that can affect data heterogeneity.

Hsieh S.Y. et al. used magnetic bead-based MALDI-TOF MS to systematically assess the impact of sampling, processing, and storage on the proteomic profile of low-molecular-weight components in serum and plasma samples. It was found that sampling procedures, including the selection of tubes for blood collection and the respective anticoagulants, changes in coagulation time, and the time delay before centrifugation and hemolysis, have a significant effect on the proteomic composition of low-molecular-weight components in biosamples. Moreover, the study demonstrated that serum and plasma were mutually incompatible for a comparative analysis of proteomic composition. Based on the analysis of approximately 100 peaks, the researchers concluded that the particularities of patients’ food intake, protocols of the pre-analytical stage (speed and time of centrifugation), storage conditions (at 4 or 25 °C for 24 h or at −80 °C for 3 months), and repeated freezing/thawing for up to 10 cycles had a relatively insignificant effect on the proteomic composition of biosamples. However, the type of sample, the processing, and the storage had an effect on the low-molecular-weight components of serum and blood plasma [53].

Therefore, if different standard operating procedures are used, the comparison of results becomes difficult. These aspects may be potential barriers to achieving the overall goal of an international research structure aimed at facilitating access to standardized human biological materials. For this reason, it is necessary to ensure uniformity of at least the basic standard operating procedures through guidelines for the widespread and efficient use of biomaterials [75].

Such guidelines are developed and updated by organizations such as ISBER [22], the European and Middle Eastern Society for Biopreservation and Biobanking (ESBB) and the Biobanking and BioMolecular Resources Research Infrastructure-European Research Infrastructure Consortium (BBMRI-ERIC) [76].

## 6. Ethical Aspects of Human Biobanks

Biobanks have given rise to a new field called biobanking. This term covers not only the biobank infrastructure itself, but also sample collection, biosample data, engineering, and technological solutions. The term “biobanking” also refers to the whole range of social, legal, and ethical problems that must be resolved as biobanks develop. For example, any biobank is interested in the fact that a citizen, voluntarily and at the expense of informed consent, transfers his/her material to the biobank, without strictly determining the purpose for which the material is to be used. If a citizen has stipulated conditions—for example, has identified a range of scientific problems in which he/she is not ready to participate through his/her genetic material, then the biobank is faced with the task of tracking the access to the pool of samples and materials, allowing only for specific studies to make use of the samples of particular citizens. This creates organizational, technological, and financial difficulties. In addition, a citizen may withdraw his/her consent at the stage when the obtained results are to be published or during the processing of his genetic information.

However, how to link the autonomous right of each person to dispose of their genetic material, including the right to determine what types of scientific research are permissible, with the task of gaining mass knowledge is a major problem for biobanks. A problem of this kind cannot be solved via a standard algorithm. Therefore, each biobank, depending on its affiliation, funding format, and collection scale, is constantly forced to address issues related to ethical and legal regulation.

*Informed consent* is one of the most discussed topics in the context of biobank research. The goal is to enable a person to decide whether to participate in a research program or not. Informed consent is considered an ethical and legal document that protects the rights of participants and/or patients and maintains public trust [77].

In Europe, the General Data Protection Regulation (GDPR-2018) has formed a regulatory framework containing principles for the collection and processing of personal data of individuals within the European Union.

However, despite the recommendations of GDPR-2018, the issue of optimizing consent for various kinds of studies within the framework of biobanking remains controversial [78], because there are some limitations [79]. The main limitations are the amount of information and the complexity of participant perception [80]. For example, do participants understand the purpose of the study and the risks stated in informed consent, or do they understand that their participation is voluntary and that they have the right to refuse? Regarding informed consent, it is important that potential study participants understand what their consent entails to the greatest extent possible [80].

Nishimura et al. showed that enhanced forms of consent were among the most effective [81]. This model of “broad agreement” is used by most biobanks nowadays [78]. The basis for such consent is an agreement on the use of donor biomaterial for current or future research without the need for contact with the patient. This provides researchers with the flexibility to conduct a wide range of scientific studies. However, it is implied that patients agree that they may not receive information about the results of the study in which their biomaterial was used. The use of another type of consent, dynamic consent, requires the availability of tools for easy access and constant contact with the patient, which makes it possible to obtain biomaterial from the patient for each new study [82]. However, additional research is required to create the optimal unified informed consent. In order to assess how clear a certain form of informed consent is to the patient (donor of the biomaterial), it is important to conduct the relevant bioethical research. This will allow us to evaluate the understanding and stance of donors, use the results to adapt the informed consent, and generate a new document that is understandable by the donor [83].

Bossert et al. investigated informed consent patterns through interviews in 11 focus groups. It was concluded that, in order to improve the quality of information perception (IP), existing forms of IP require the adjustments described below [82].

(1)The main page of the IP should contain the most important aspects, such as the purpose of the document, a description of the biobank, and the rights of participants (for example, the opportunity not to participate in the study or to refuse to participate at any time, etc.).(2)The main section of the document should contain information on the nature and functioning of the biobank, the collection and storage of data, the rights of participants, etc.(3)The process of biobank donation and further research should be illustrated.(4)Information about the concepts of “long-term storage”, “random data relevant to your health”, and the ability to communicate with the participants should be more clearly and thoroughly explained.(5)Sentences that are too long and lengthy wordings should be deleted.(6)Definitions of technical terms, such as “biomaterials”, “Research ethics committee”, “pseudonymization”, and so on, should be the most accurate and understandable [84].

*Protecting the personal data* of the research participants is the paramount ethical issues [83]. The first step in the data securing is correctly executed informed consent [84]. Biobanks typically store personal information related to a particular phenotype. These data constitute a serious threat to personal life [85,86,87]. The fears of the study participants are due to their personal data may become available to insurance companies and employers, therefore biobanks must guarantee personal data security.

In addition, research results can also negatively affect not only individuals, but entire groups that may feel stigmatized, for example, because of their genetic predisposition. Biobanks that conduct research on a particular ethnic or other specific group of people should take this into account when publishing research results. Therefore, ethical issues of data security and privacy must be configured in all aspects of the biobank organization [88,89].

The usage of anonymization for biosamples is the best way to protect personal information, but it limits the practicality of research because many biobanks cannot sufficiently use this data [90,91]. For example, related information about the genome and phenotype may be lost. Likewise, there is no feedback opportunity from participants. Therefore, many biobanks refuse anonymity and prefer to encode information as the most amenable way to ensure privacy: simple, double or triple encoding is acceptable in standard research practice and at the same time safe enough to provide a satisfactory option of personal data protection [91,92,93].

Kaufman D.J. showed that the protection of personal data is a significant issue for nearly 90% of subjects [94]. The consequences of privacy breaching can significantly affect the public’s willingness to participate in research. Therefore, biobanks should always guarantee the maximum level of protection for participants’ personal data [83].

*Participants access to research results:* research results may represent statistical, clinical, and/or investigation interests, but also researchers often get random results. A common ethical standard requires that everyone has the opportunity to choose if they want to know results of the study.

The issue of returning research results to participants is relevant. Few literary data show that participants expressed significant interest in obtaining research results that used their biomaterial [95,96]. However, most authors oppose the return of individual results [83,90,97,98,99]. In their opinion, the results of the study can be misinterpreted, and consequently, stipulate concern among participants, especially if the information does not have clinical significance. Therefore, the authors adhere to the position that the return of the result is flexible only with a very high clinical significance.

The main goal of research associated with the activities of biobanks is to expand information that in the long run will lead to a cooperative improvement in the quality of life and improve people’s health.

One of the important conditions for the effective conduct of research associated with the activities of biobanks is to *ensure public confidence*. A lack of public confidence can lead to adverse effects on the biobanking structure [97]. Having studied public opinion about the activities of biobanks, the authors came to the conclusion that the majority of surveyed people are positive and flexible to take part in research [94,100,101]. At the same time, few of them understand the real goal of research and biobanking; however, they are enthusiastic about it and show a desire to participate [102,103].

To date, the work of biobanks is successful and supported by society. However, biobanks should continue to increase public confidence. Consent and protection of personal information are essential elements of this process [104,105].

An important ethical issue is *the participation of minors in biobank research*. Minors cannot fully go bankrupt in ethical and other issues. Therefore, most biobanks do not attract this category to the study due to increased attention from society and the media. However, this may lead to a delay in medical studies of children on studies of adults. From an ethical point of view, in the end, underage children will suffer relatively more than adults. Most authors support the idea of involving children in biobank research, but they also agree that their risk should be minimized [106,107,108,109]. Some authors suppose a delay to the exchange of study results until the participating children reach adulthood [110]. Other authors oppose this view, arguing that such a proposal may seriously delay research, and may leave an entire generation behind [111,112,113].

The parents of minor children are interested in returning the results of studies in which their children participated, but some authors believe that it is not ethical [109,114].

Despite the fact that parents have the right to decide whether to involve their minor children in the study or not, whether to give informed consent in their place, children must decide whether they want, upon reaching adulthood, to find out the results of studies they contributed in.

In some cases, the study may be conducted without the consent of the parent or child. These situations are rare, but the organizers of the study should provide minimal risk or complete lack of risk for children. Researchers should always respect the fears and/or disagreement of a child to be involved in. In each study conducted with children, ethical advice should be consulted to protect the concerns of child [115,116].

In addition, a special level of protection should be provided for those who suffer mental illness. In this case, the guardian has the opportunity to sign an informed consent, and in all such cases it is necessary to consult with ethical advice to protect the interests of such volunteers [98].

Another topic discussed in the literature is *ownership of biological samples and data*. This issue is ethically and legally challenging in the biobanking structure. It is believed that the legal position regarding property remains unresolved [98]. Other authors believe that complete anonymization will actually make biological materials unclaimed, but in all other cases, donors retain ownership and will be able to withdraw their consent and their biological material transferred to the biobank [117,118].

Most biobanks support the idea of being custodians or trustees of biosamples, but not their owners [119,120,121]. O’Brien S.J. suggested that the biological material of donors should be the common property of donors and researchers [91], but this issue remains relevant and open for discussion.

Biobanks are expensive infrastructures that require highly qualified personnel, modern equipment, its maintenance, and a large number of consumables [82]. To maintain stability, some biobanks function as business units and use the financial potential of their samples and data. However, this can cause ethical and legal problems [122]. International biobanks have agreed that stored human biosamples cannot be used as commercial products [123].

Several biobanks have developed self-financing strategies. For example, the UCSF AIDS Specimen Bank (https://cfar.ucsf.edu/cores/aids-specimen-bank) has developed a cash replenishment methodology to recover costs associated with sample processing, storage, consumables, software and hardware maintenance, data management, and data exchange.

## 7. Conclusions

Currently, there is a growing demand for the accurately annotated samples/specimens with trackable clinical history which may improve full-fledged research [123]. Long-term sample collection, storage and monitoring are consuming tasks requiring expensive infrastructure and well-established collaboration with clinics as the main source for samples. 

Biobanking is a new and innovative field of research which is organized in a strict workflow manner. In general, biobanks must be evaluated as a professional organization for biomaterial storage, handling and research management. Despite the involvement of international organizations such as BBMRI-ERIC (www.bbmri-eric.eu/) and ISBER (www.isber.org) and government agencies in for the use of repository resources [17,124], heterogeneity in samples collection and storage, ethical and legal issues related to access the samples (especially, when the material obtained from children), personal information safety, consent limitations and data management for research purpose are still challenging tasks [125]. Therefore, the incorporation of the efficient IT technologies, standards implementation, compliance with the international standards for samples and data custody are matters that merit paramount attention.

Recently, the scientific community has delivered the need for organization of the targeted biobanks which may now be actively involved in research projects and government programs. Their tasks include: (1) the receipt, processing, and storage of samples of biological origin; (2) the ability to create “added value” of biological samples; (3) the distribution, at the request of competent institutions, of materials for scientific purposes, etc.

Interest in usage of a wide variety samples, accessibility of collections for research purposes throughout the scientific communities are the most promising ways to improve and rapidly develop the biobanking interface. Presumably, the ongoing interest to personalized custom-tailored treatment can also prove the significance of biobanks. It seems on the surface that the main concept of biobanks is collecting, storing, sorting and preservation of biological sample for the long-term scientific purposes unlike clinics and hospitals which are, primarily, designed for diagnosis and are very limited for samples management.

However, the primary role of biobanks is not merely collecting, storage, and intended use of specimens, but make it in strict association with relevant clinical data; otherwise, the collected specimens are beyond scientific value. In this regard, biobanks take the role of a personal data guardian by way of their integrity, thus, state-of-the-art infrastructure, including electronic medical records, becomes a significant side of biobanks operation.

The ability to compare data and biological samples from different biobanks is critical to speeding up the pace of translational research. A meta-model for describing biobank information is already under development as a first step in exchanging data between biobanks that exhibit huge heterogeneities. This work is carried out internationally to help harmonize the national biobanks. Information on participating biobanks is collected by a common set of attributes (minimum data set) intended for the adoption of various types of collections. The latest interaction and semantic network technologies can be used to build resource description structures for data and services, providing flexible structures that can be used in various data sharing scenarios.

## Figures and Tables

**Figure 1 diagnostics-10-00485-f001:**
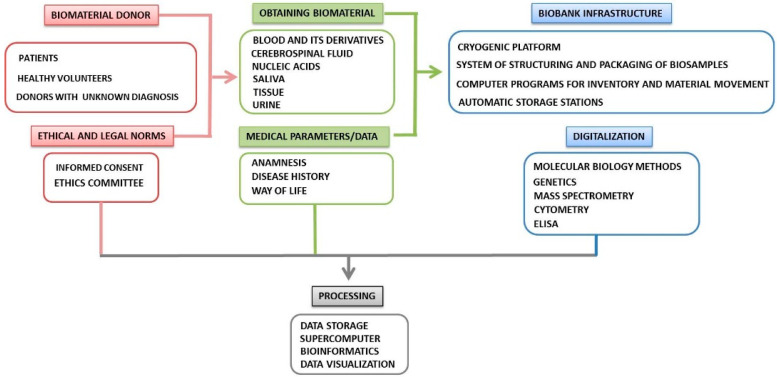
Key components of biobanking and routing scheme of biological samples.

**Table 1 diagnostics-10-00485-t001:** CEN technical specifications recommended for in vitro molecular diagnostic studies.

No.	ISO	Title	Ref.
1	ISO 20184-1:2018	Specifications for pre-examination processes for snap frozen tissue. Part 1: Isolated RNA	[27]
2	ISO 20184-2:2018	Specifications for pre-examination processes for snap frozen tissue. Part 2: Isolated proteins	[28]
3	ISO 20166-1:2018	Specifications for pre-examination processes for FFPE tissue. Part 1: Isolated RNA	[29]
4	ISO 20166-2:2018	Specifications for pre-examination processes for FFPE tissue. Part 2: Isolated proteins	[30]
5	ISO 20166-3:2018	Specifications for pre-examination processes for FFPE tissue. Part 3: Isolated DNA	[31]
6	ISO 20186-1:2019	Specifications for pre-examination processes for venous whole blood. Part 1: Isolated cellular RNA	[32]
7	ISO 20186-2:2019	Specifications for pre-examination processes for venous whole blood. Part 2: Isolated genomic DNA	[33]
8	ISO 20186-3:2019	Specifications for pre-examination processes for venous whole blood. Part 3: Isolated circulating cell free DNA from plasma	[34]

ISO—International Organization for Standardization; FFPE—formalin-fixed, paraffin-embedded; TS—Technical Specification.

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
