# Peer review of "Biobanks—A Platform for Scientific and Biomedical Research"

_diagnostics, 2020, doi:10.3390/diagnostics10070485_

Round 1
Reviewer 1 Report
Very difficult to make recommendations because the lines of the manuscript are not numbered.
At the final of the Introduction section please highlight the aim of this research and the novelty/special aspects it brings in the field.
Please revise the entire manuscript regarding References missing. All the standards/groups of work/etc. need References to be added. Furthermore, all the abbreviation must to be explained at first mentioning in the text. see bellow few examples:
"The controlled, highly qualified organization and operation of storage facilities and/or infrastructures for biomedical research is the ultimate goal of biobanking. In this context, the ISO/Technical Committee (TC) 2761/Working Group (WG) 2 Biobanks and Bioresources was formed to “elaborate a package of international Standards in the Biobanking field, including human, animal, plant, and microorganism resources for research and development, but excluding therapeutic products” [references needed]. The use of ISO/DIS 20,387 is applicable to “all organizations performing biobanking activities, including biobanking of human, animal, plant, and microorganism resources for research and development” [reference needed]. Compliance with these requirements will enable biobanks to work consistently, professionally, and provide biological resources of adequate quality."
"The European Committee for Standardization regularly conducts numerous activities based on the results of different initiatives (please explain the abbreviation including SPIDA) [reference needed] to develop new technical specifications for International Organization for Standardization (ISO), scopes for the standardization of pre-analytical workflow, best transfer procedures and quality assurance in molecular analysis of different types of samples.
The ISO-15189 standard [reference needed], based on the ISO/International Electrotechnical Commission (IEC) 17,025 and ISO-9001 [reference needed], provides general competency requirements for testing and calibration laboratories and a quality management system (QMS)."
In this regard, to Table 1 one more column must be added, dedicated to references for each Technical specification.
In the paragraph:
"Protecting the personal data of the research participants is the paramount ethical issues. [51].space neededThe first step in the data securing is correctly executed informed consent [52]. Biobanks typically store personal information related to a particular phenotype. These data constitute a serious threat to personal life."
Please avoid to use paragraphs of 1 sentence/phrase. Conceptually is incorrect and it interrupts the flowing of reading the paper. Use separate paragraphs just for different ideas.
References section: please revise - all references are numbered twice. In the manuscript, all the references must to be located before the final point of the sentence/phrase, NOT between 2 points as can be seen above, highlighted on blue background.
Author Response
Dear Editor,
our reviewer response is attached.
Kind regards,
Аuthors
Reviewer 2 Report
The manuscript is interesting and well structured.
However we have minor suggestions to present:
- In the "Pre-Analytical Sample Handling" section, the authors cite ISO-15189 as the reference standard. They should update the information based on the new ISO 20387: 2018 standard.
- After the paragraph "Cell Banks", a paragraph on organoid biobanks with the corresponding references should be produced.
- Finally, they should discuss the possibility of integrating biobanks with data from genetic studies carried out on biomaterials. This would enrich the development potential of biobanks also for diagnostic / therapeutic purposes.
Author Response

(The authors gave the same response as above.)

Reviewer 3 Report
Biobanks – A Platform for Scientific and Biomedical Research
The title does not correspond the article correctly.
The definition of biobank in the manuscript does not meet the current knowledge about biobanks, biobanks are not only collections of material, but also matched information. Biobanks were defined by OECD, ISBER, BBMRI-ERIC, ESBB, and also European Commission, so the definition is not vague.
The classification of biobanks is not complete, the purpose of human biobanks is not fully explained, the importance of data, their structure, volume, diversity and the tools how to manage biobanking data is not sufficient.
As regards references, main literature published in the ISBER Journal Biopreservation and Biobanking, that covers the topic is missing, as well as the main documents of ISBER, BBMRI-ERIC, and ESBB. Biobanks are not only samples; biobanks are both samples and data. The description of current status of biobanks and biobanking science is not complete and fully actual.
In the chapter „Organization of biobank“ the content is not well balanced with accent on pre-analytic methods, and standards, for inspiration I recommend ISBER Best Practices.
Chapter 3. „Pre-analytical sample handling“ only general information that does not describe pre-analytics for all types of materials in biobanks, in general better to use ISBER Best Practices.
Chapter 4. „Collection and Storage of Biomaterial“, is not balanced between types of material, methods., 4.1, 4.2, 4.3, why were only these three types of material chosen? More common are biobanks collecting FFPE samples, blood, serum, plasma. New modern and perspective and e.g. microbiom biobanks.
Chapter 5. „The heterogeneity of Biobanks“ , here ISBER activities worldwide, ESBB and BBMRI-ERIC initiatives should be mentioned.
Chapter 6. „Ethical Aspects of Human Biobanks“, is so complicated topic, that require a special attention, e.g. how many „informed consents“ are being currently used, anonymization versus pseudonymization, should be explained, the topic is more complex. Children biobanks are very sensitive topic, the characteristics of biobanks as infrastructures is not explained satisfactory.
Chapter 7. „Conclusions“, does not answer the title of the manuscript.
General comments:
As regards references, main literature published in the ISBER Journal Biopreservation and Biobanking, that covers the topic is missing, as well as the main documents of ISBER, BBMRI-ERIC, and ESBB. Biobanks are not only samples; biobanks are both samples and data. The description of current status of biobanks and biobanking science is not complete and fully actual.
Author Response

(The authors gave the same response as above.)

Round 2
Reviewer 3 Report
the authors had partly done the corrections based on previous review report.
I recommend:
in Introduction to mention definition of biobank by OECD, ISBER, EC (European Commission), these are institutions, not only projects like P3G
chapter 4: to mention the classification of biobanks as: population based biobank, disease oriented biobanks and virtual biobanks, not only based on material stored in biobank
additional recommendations: not "bespoke" but "custom -tailored" (page 3, 624)
line 208 is "SPIDA" ? "SPIDIA"
BBMRI-ERIC is not explained in the text Lines 457, 610
in Conclusions, must be mentioned the purpose of biobanks not only samples, but also "DATA", because samples without connected data are of no scientific use!
Author Response
We have changed the content of the manuscript in accordance with the recommendations of the reviewer. Changes made to the manuscript are highlighted in green.
Sincerely, the authors
Dr. Malsagova Kristina
Question 1. In Introduction to mention definition of biobank by OECD, ISBER, EC (European Commission), these are institutions, not only projects like P3G
Answer:
Thanks for the recommendation, we have added the definition of “Introduction” to the biobank from the OECD, ISBER, EC (band 35-45).
Question 2. chapter 4: Mention the classification of biobanks as: population based biobank, disease oriented biobanks and virtual biobanks, not only based on material stored in biobank
Answer:
Thank you, we took into account the comments, we updated information about population biobank to the section “Collection and storage of biomaterial” (lane 253-270).
Question 3. Additional recommendations: not "bespoke" but "custom -tailored" (page 3, 624)
Answer:
We agree with the remark and made an appropriate change to the text (lane 663).
Question 4. line 208 is "SPIDA" ? "SPIDIA"
Answer:
Thanks for the comment, we fixed the typo (lane 219).
Question 5. BBMRI-ERIC is not explained in the text Lines 457, 610
Answer:
We decoded the abbreviation BBMRI-ERIC at the first mention in the manuscript (lane 490).
Question 6. in Conclusions, must be mentioned the purpose of biobanks not only samples, but also "DATA", because samples without connected data are of no scientific use!
Answer:
We have supplemented the “Conclusion” section in accordance with the reviewer's recommendations (lane 668-680).